# Cross-Border Document Service Procedures in the EU from the Perspective of Italian Practitioners—The Lessons Learnt and the Process of Digitalisation of the Procedure through e-CODEX

Rosanna Amato *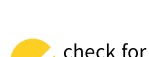 and Marco Velicogna *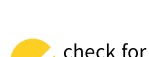

Institute of Legal Informatics and Judicial Systems of the National Research Council of Italy, Bologna Branch, 40126 Bologna, Italy
* Correspondence: rosanna.amato@bo.igsg.cnr.it (R.A.); marco.velicogna@cnr.it (M.V.)

**Abstract:** An effective legal framework for judicial cooperation in the field of the service of documents is a keystone for the effective functioning of the area of freedom, security and justice, as referred to in the Treaty on the EU. In particular, the proper service of a claim to the addressee is a necessary step for starting a proceeding and, simultaneously, an essential requirement for exercising the right of defence. The EU has adopted specific provisions to remodel the traditional channel of documents' transmission with smoother solutions that assist cross-border judicial proceedings. Despite this, the European service procedure is not that straightforward and can still be very complex for most users, causing additional costs and legal uncertainty. Against this background, this article explores how the cross-border service of documents works in practice. It presents the findings resulting from empirical exploratory research carried out in Italy to assess the concrete use and usability of the European rules adopted to simplify, speed up and reduce the costs of cross-border service of judicial and extrajudicial documents in civil and commercial matters, also in the view to support a possible digitalisation of the procedure. Building on empirical data, the paper brings to light the existing hiatus between the service procedure 'on the books' and the reality of how the relevant provisions are applied daily, so as to provide solid ground for reflecting on the current situation and on the impact that the recast Regulation 2020/1784/EU, which took effect in July 2022, will have to the supranational system of cross-border service of documents, in particular concerning the potential of the use of ICT to support it.

**Keywords:** service of documents; judicial cooperation in civil matters; cross-border notification; cross-border communication; digitalisation of cross-border procedures; EU law

## 1. Introduction

Cross-border social and commercial interactions have increased sharply in Europe in the past few years. This increase is due to various factors, ranging from the greater mobility of the new generation of workers to the rise of international e-commerce. For example, based on the Eurostat statistics, in 2021, nearly 74% of internet users living in the EU bought goods or services online, and 32% of these individuals ordered their purchases from sellers based in the other Member States.[1] While the coronavirus pandemic has temporarily reduced the movement of people, it has also "brought an unforeseen acceleration of the use of digital services worldwide",[2] allowing for remote working and schooling, but also

---

1   See Eurostat Statistics Explained, E-commerce statistics for individuals. Available online: https://ec.europa.eu/eurostat/statistics-explained/index.php?title=E-commerce_statistics_for_individuals#General_overview (accessed on 30 June 2020).
2   See The effects of digital technology during the COVID-19 pandemic, 12 June 2020. Available online: https://www.ecommerce-europe.eu/news-item/the-effects-of-digital-technology-during-the-covid-19-pandemic/ (accessed on 30 June 2020).

pushing the population to increasingly buy online products and services (Cruz and Dias 2020), accelerating the decoupling of these activities from the physical geography.

Such an escalation in the number of cross-border transactions, together with the increased EU mobility (primarily in the context of tourism and study abroad), is undoubtedly one of the indicators that the internal market is functioning smoothly. However, this is also one of the reasons explaining the increase in the number of cross-border disputes. On this point, the European Commission reports that every year there are approximately 3.4 million civil and commercial court proceedings in the EU having cross-border implications (European Commission 2018). This figure should be added to the number of out-of-court proceedings and those situations that are not reported by wronged citizens, who are deterred by the lack of understanding of cross-border justice, its cost and its complexity.

It follows that, within this context, developing an effective European area of justice in civil matters and providing an efficient framework for cross-border judicial cooperation has become decisive (Mc Clean 2002; Dominelli 2018),[3] both for the proper functioning of the internal market and, more in general, for supporting trust in cross-border situations. This is all the more crucial as it directly impacts how people "perceive the functioning of the judiciary and the rule of law in the Member States".[4] In this respect, several European legal instruments have been adopted to remedy the challenges of an increasingly integrated cross-border society. Innovative mechanisms have been put in place to enhance cooperation and provide better access to justice throughout the EU.[5]

As part of this effort, the EU legislator has adopted specific provisions to smooth cross-border service of judicial and extrajudicial documents in civil and commercial matters, while safeguarding a high level of security in the transmission of legal documents and the rights of the addressee. The legislation covering this topic tackles an issue that has an impact on the daily lives of EU citizens and businesses. The proper service of a claim to the addressee is a necessary step for starting a proceeding and protecting the claimant's legitimate expectations, thus avoiding paralysing the judicial system. At the same time, it is an essential requirement for exercising the right of defence of the addressee as enshrined by the EU Charter of the Fundamental Rights[6] and national constitutions. For this reason, over

---

[3] See also; the Proposal for a Regulation of the European Parliament and of the Council amending Regulation (EC) No 1393/2007 of the European Parliament and of the Council on the service in the Member States of judicial and extrajudicial documents in civil or commercial matters (service of document) Brussels, 31 May 2018, COM(2018) 379 final, 2018/0204(COD); Commission Staff working documents evaluation, accompanying the document proposal for a Regulation of the European Parliament and of the Council amending Regulation (EC) n. 1373/2007, 2018, p. 12.

[4] Proposal for a Regulation of the European Parliament and of the Council amending Council Regulation (EC) No 1206/2001 of 28 May 2001 on cooperation between the courts of the Member States in the taking of evidence in civil or commercial matters Brussels, 31 May 2018. COM(2018) 378 final, 2018/0203(COD).

[5] The European action in the field of judicial cooperation in civil matters is vast and varied, also covering crucial matters, such as family law or rules for non-contractual obligations. Overall, specific legal instruments have been adopted to ease the determination of jurisdiction and the recognition of decisions in extra-judicial cases (Regulation (EU) No 1215/2012 on jurisdiction and the recognition and enforcement of judgments in civil and commercial matters; Council Regulation (EC) No 2201/2003 concerning jurisdiction and the recognition and enforcement of judgments in matrimonial matters and the matters of parental responsibility, repealing Regulation (EC) No 1347/2000; Council Regulation (EC) No 4/2009 on jurisdiction, applicable law, recognition and enforcement of decisions and cooperation in matters relating to maintenance obligations; Regulation (EC) No 805/2004 creating a European Enforcement Order for uncontested claims; Regulation (EU) No 650/2012 on jurisdiction, applicable law, recognition and enforcement of decisions and acceptance and enforcement of authentic instruments in matters of succession and on the creation of a European Certificate of Succession), to harmonise conflict of law rules (Regulation (EC) No 593/2008 (Rome I) and the Regulation (EC) No 864/2007 (Rome II) seek to improve the legal certainty and predictability of the outcome of litigations concerning non-contractual obligations. Together with this, the Regulation (EU) N. 1259/2012 they establish a comprehensive legal framework for divorce and legal separation), as well as to facilitate access to justice (Council Directive 2002/8/EC of 27 January 2003 to improve access to justice in cross-border disputes by establishing minimum common rules relating to legal aid for such disputes; the Regulation (EC) No 861/2007 of the European Parliament and of the Council of 11 July 2007 establishing a European Small Claims Procedure; Regulation (EC) No 1896/2006 of 21 May 2008 on certain aspects of mediation in civil and commercial matters) and smooth cross border cooperation between civil courts (e.g., Regulation 1206/2001 adopted to simplify and expedite judicial cooperation in taking of evidence in civil matters).

[6] Charter of the Fundamental Rights of the European Union, art. 47.

the years, the relevant EU legal framework has been remodelled to replace the traditional (and more cumbersome) transmission channel of documents based on international conventions with simpler and faster systems for ensuring the efficient cross-border exchange of judicial and extrajudicial documents for the purpose of service between the Member States.[7] Most notably, an innovative transmission channel based on the direct forwarding of documents for service between decentralised national (transmitting and receiving) agencies has been established to make this workflow smoother, faster and more secure.[8] To date, this mechanism can be regarded as the main step forward in judicial cooperation between Member States in the field of transmission of documents for service; nevertheless, it can still be very complex for most users. Evaluations conducted over the years have revealed that—while providing many advantages compared to the previous legal regime—this system is still underperforming. It works more slowly and less efficiently than expected and the proposed deadlines are not regularly met. What has emerged is that gaps in the legal framework and in the way judicial and extrajudicial documents are serviced lead to delays and costs for citizens, businesses, and public administrations. Limits in the service of documents regulation also result in shortcomings in the protection of procedural rights and increase the overall legal complexity and uncertainty of cross-border judicial procedures.[9]

In light of the reported sub-optimal performance, a further process of regulatory reform was initiated in 2018, which has led to the adoption of a new set of rules aimed

---

[7]　This matter was firstly addressed with the Council Regulation 1348/2000/EC on the service in the Member States of judicial and extrajudicial documents in civil or commercial matters, and then governed by the Regulation 1393/2007/EC on the service in the Member States of judicial and extrajudicial documents in civil or commercial matters (service of documents). Regulation 1393/2007 has recently been replaced by the Regulation 2020/1784/EU on the service in the Member States of judicial and extrajudicial documents in civil or commercial matters (service of documents) (recast), which has entered into force on 1 July 2022.

[8]　See Proposal for a Council Directive on the service in the Member States of judicial and extrajudicial documents in civil or commercial matters, COM(1999) 219 final, 99/0102 (CNS), p. 11. This channel was first introduced by the Council Regulation 1348/2000/EC, art. 2, then replaced by Regulation 1393/2007/EC, art. 4 and is now governed by Regulation 2020/1784/EU, art. 3.

[9]　The most recent ex-post evaluation report of the Regulation (EC) No 1393/2007, which shed light on the limitations that still hinder the functioning of the European system of cross-border service of documents, was made public in 2018 and formed the basis for the Impact Assessment Accompanying the document Proposal for a Regulation of the European Parliament and of the Council amending Regulation (EC) No 1393/2007 (SWD(2018) 287 final). The problem definition in the Impact Assessment accompanying the Proposal was also based on the findings of the regulatory fitness (REFIT) evaluation undertook by the Commission to assess the operation of the instrument in relation to the five key mandatory evaluation criteria of effectiveness, efficiency, relevance, coherence and EU added value. Before then, the following studies and reports by the EU Commission (or commissioned by the Commission) have assessed the implementation of the Regulation: (a) European Commission, Directorate-General for Justice and Consumers, Gascón Inchausti, M., B. Hess, G. Cuniberti, et al. An evaluation study of national procedural laws and practices in terms of their impact on the free circulation of judgments and on the equivalence and effectiveness of the procedural protection of consumers under EU consumer law: strand 1: mutual trust and free circulation of judgments, Publications Office, 2017. Available online: https://data.europa.eu/doi/10.2838/38491 (accessed on 15 September 2022); (b) European Commission. Directorate-General for Justice and Consumers, Simoni, A., Pailli G., Study on the service of documents, Comparative legal analysis of the relevant laws and practices of the Member States—Final Report, No JUST/2014/JCOO/PR/CIVI/0049 5th October 2016; European Commission. 2014. Directorate-General for Justice, Study on the application of Council Regulation (EC) No 1393/2007 on the service of judicial and extra judicial documents in civil or commercial matters: final report, Publications Office. Available online: https://data.europa.eu/doi/10.2838/84790 (accessed on 15 September 2022); (c) Report of the Commission on the application of Regulation (EC) No 1393/2007 of the European Parliament and of the Council on the service in the Member States of judicial and extrajudicial documents in civil or commercial matters (service of documents)—COM(2013) 858 final; DMI in Consortium with University of Florence and University of Uppsala, Study on the service of documents. Comparative legal analysis of the relevant laws and practices of the Member States. Final Report (No JUST/2014/JCOO/PR/CIVI/0049), 5 October 2016; MainStrat, Study on the application of Council Regulation (EC) No 1393/2007 on the service of judicial and extra judicial documents in civil or commercial matters, report realised upon request of the EU Commission, 2014, ISBN: 978-92-79-34791-7. Together with this, the European Judicial Network in civil and commercial matters has dedicated many of its meetings to the evaluation of the application of the Service Regulation e.g., (15–16 May 2014; 2 October 2014; 14–15 November 2016; 30 November–1 December 2017). Problems resulting from the application of the relevant legislation have been also highlighted in the Proposal for a Regulation of the European Parliament and of the Council amending Regulation (EC) No 1393/2007 of the European Parliament and of the Council on the service in the Member States of judicial and extrajudicial documents in civil or commercial matters (service of documents), 2018/0204(COD).

at improving the effectiveness and speed of judicial procedures, primarily by digitalising them. In line with the latest e-justice EU policies,[10] the new provisions take digital communication a step further by obliging the competent authorities of the Member States to communicate with each other using e-CODEX (Kramer 2022), which is a decentralised and interoperable IT system, created to enable the digital exchange of case data in cross-border legal proceedings.[11] As it will be better explained below (paragraph 5), e-CODEX is a technological innovation that is supposed to fundamentally change the way the judiciary works in cross-border procedures, as it provides for the dematerialisation of judicial proceedings and communication between judicial authorities.[12] After more than a decade, this project has firmly established itself, so much so that it features prominently in the recast regulation, where it represents the main change.

As in many other areas in which communication and transmission of data and documents is required, technology can indeed provide great help in addressing existing shortcomings. Yet, the point we will try to explain in this article is that despite the indisputable enhancing role the use of technology will certainly have on the practical use of this cooperation procedure, it cannot provide a solution on its own. As experience has shown (Velicogna et al. 2015; Velicogna 2022), even if the legislation allows and supports technology use in legal proceedings (Contini and Cordella 2016; Velicogna 2019) and adopts a 'technology-neutral' language, cross-border ICT solutions are not easy to implement. First, to function, they need the will of the Member States to participate. Also, even where such intention to participate is established, legal obstacles, lack of cross-border interoperability of the national ICT systems, and priority given to domestic needs add up to increase the difficulty of implementing a viable technological solution. The question, though, is not limited to the need to authorise the use of technology. The procedure's legal framework must be aligned with the technological design possibilities. Attempts of isomorphic translation of offline procedures into digital ones have a long tradition of failures in the justice domain (Contini and Fabri 2003; Velicogna 2007; Schmidt and Zhang 2019; Carnevali 2019). Technological solutions, which often appear simple to use and are perceived as tools that make people's lives easier in everyday settings, turn out to be complicated when they have to be integrated into judicial services, which are highly regulated and have been generally designed to be carried out offline (Contini and Fabri 2003).

On this assumption, the main aim of this article is to provide a deeper understanding of how the cross-border service of documents works in practice within the EU legal landscape and the role ICT can actually have in supporting it. The findings presented therein result from exploratory research carried out by the Italian team within an EU co-financed project (Me-CODEX),[13] which was meant to ensure a rapid and sustainable transition to the long-term sustainability of the e-CODEX digital infrastructure. As part of the Me-CODEX project, the Italian research team has been in charge of assessing the concrete use and usability of the European rules adopted to simplify, speed up and reduce the costs of cross-border service of judicial and extrajudicial documents in civil and commercial matters, in the

---

10  See: Communication on the digitalisation of justice in the EU in December 2020 (JOIN/2020/18 final); Commission work plan for 2021 as a 'digital judicial cooperation' package (COM/2020/690 final); The 2021 EU Justice Scoreboard, Communication from the Commission to the European Parliament, the Council, the European Central Bank, the European Economic and Social Committee and the Committee of the Regions COM(2021) 389.

11  At present, the cross-border electronic exchange of data in the area of judicial cooperation in civil and criminal matters by means of the e-CODEX system i is governed by Regulation (EU) 2022/850 of the European Parliament and of the Council of 30 May 2022 on a computerised system for the cross-border electronic exchange of data in the area of judicial cooperation in civil and criminal matters (e-CODEX system), and amending Regulation (EU) 2018/1726.

12  This resulted from a EU-funded multiannual project carried out between 2010 and 2016, and since then, it has undergone a series of follow-up projects aimed at maintaining it (Me-CODEX), expanding its use to other procedures and an increasing number of Member States (EXEC, IRI, e-CODEX Plus, CEF e-Justice DSI) as well as opening up this infrastructure to third parties or the legal professions, so as to provide services that meet the expectations and capacities of litigants and other stakeholders (Pro-CODEX, API for Justice).

13  "Me-CODEX: Maintenance of e-Justice Communication via Online Data Exchange", JUST/CEF-TC-2018-CSP-ECODEX.

view to support a possible complete digitalisation of the procedure through e-CODEX and, therefore, allowing one to transmit an electronic decision smoothly.[14] This article builds on the results obtained in the course of this research.

After having placed this work in the relevant literature and described the methodology used (paragraph 2), this article will first describe how the dialogue between national authorities works—or should work—on paper (paragraph 3). Then, in order to go into greater detail on this subject, avoid redundancy, and contribute to the relevant lines of research currently ongoing, it explores how the procedure actually works in the national context and how citizens, companies and practitioners have to navigate the relevant EU provisions and the differences that may occur in their application both at the national and even at a local level (paragraph 5). Building on empirical data, the existing hiatus between the cross-border service procedure "*on the books*" and the reality of how the relevant provisions are applied daily is, thus, analysed. Exploring this difference can provide a better understanding of how EU cross-border rules function in national settings, as, in practice, the application of transnational cooperation procedures is the result of local interactions based on local understandings and (often diverging) legal interpretations, as well as on stark knowledge asymmetries.

The focus is on the direct transmission method between decentralised national agencies[15] provided by the relevant EU legislation, as applied in Italy. This method is not only to be regarded as the main transmission mechanism among those available[16] but is also the method that will undergo gradual complete digitalisation through e-CODEX following the latest legal developments. Note that at the time the research was conducted and this paper was written, the EU service procedure was governed by Regulation 1393/2007/EC (hereinafter "Service Regulation"), which remained in force until the end of June 2022, when Regulation 2020/1784/EU came into effect. Therefore, the results presented in this article concern the long-term experience of practitioners during the period in which the previous Regulation was in force. However, since the system of service through decentralised agencies remains unchanged under the new legislation, this analysis can also provide a solid ground for reflecting on the changes that the new European rules will bring to the supranational system of cross-border service of documents, in particular concerning the use of ICT to support it.

## 2. Background and Methodology

The service of judicial and extrajudicial documents in civil and commercial matters is a practical necessity for effective justice (Mc Clean 2002). Questions concerning the effectiveness and efficiency of the procedures governing such transmission are, therefore, crucial and have begun to feature more and more prominently in legal doctrine with the intensification of the reform process that has affected this policy area within the EU. Since 2000, the cross-border transmission procedure for the service of judicial and extrajudicial documents in civil and commercial matters has gradually outgrown the traditional approach, based on cumbersome and slow diplomatic or consular channels that involved a series of intermediate steps until the document reached the actual addressee in the receiv-

---

[14] During the the project (2010–2016), e-CODEX has worked extensively with e-justice Service of Documents (EJS) to achieve a comprehensive overview of the legal requirements and practices for service of documents. For this reason, As legislation changes to facilitate electronic transmission of documents, this task will investigate and support the development of a complete digital procedure for claimants, which could be achieved through the implementation of an effective link between e-CODEX in-frastructure and EJS. See "e−Justice—the development of tools for secure transmission of documents by electronic means". Available online: http://www.esens.eu/sites/default/files/eJustice__the_development_of_tools_for_secure_transmission_of_documents_by_electronic_means.pdf (accessed on 9 June 2022).

[15] See the Proposal for a Regulation of the European Parliament and of the Council amending Regulation (EC) No 1393/2007 (SWD(2018) 287 final), p. 2.

[16] This method of transmission is supplemented by the following alternative systems: (a) postal service (Regulation 1393/2007/EC, art. 14); (b) direct service (Regulation 1393/2007/EC, art. 15). The Service Regulation also foresees two traditional means of interstate communication, based on the employment of diplomatic and consular channels (Regulation 1393/2007/EC, arts. 12 and 13).

ing State. Following such developments, legal doctrine has focused on this topic, mainly investigating the variety of issues related to the need to balance the drive for efficiency of supranational legislation with the respect for the fundamental rights of the parties involved, such as those related to the language regime envisaged and the procedures established to possibly refuse the document served (Gielen and Schmitz 2022; Richard 2021; Dominelli 2018; Kramer 2018; Gascón Inchausti 2017; Velicogna et al. 2015; Font i Mas 2017; Barel 2014; Galic 2013; Salvadori 2010; Dujardin 2009; Storskrubb 2008; Bohunova 2008; Franzina 2008; Daniele and Marino 2007; Adobati 2006; Biavati 2006; Frigo 2006; Carella 2004; Le-Bois 2003; Douchy 2001; Pocar 2000). More recently a new interdisciplinary strand of research has addressed the digitalisation processes of European uniform procedures in civil and commercial matters, highlighting the problems arising from the attempts to convert paper-based proceedings into digital ones. Research on the digitisation of cross-border proceedings in civil and commercial matters has mainly focused on the European Order for Payment Regulation[17] and the European Small Claim Regulation,[18] exploring both the efforts of the EU Commission to support the good functioning of these procedures through the e-Justice portal (Kramer 2016; Hess and Kramer 2017; Velicogna 2022; Onţanu 2020) and the attempt of EU Member States to create an interoperability layer allowing the interconnection of National e-Justice systems through the e-CODEX initiative (Velicogna 2014; Pangalos et al. 2014; Ontanu 2019; Velicogna 2022). These studies have explored the evolving legal framework and the solutions being set up. More recently, the exploration has also concerned the impact of digital solutions deployed during the COVID-19 pandemic (Velicogna 2021) and the next steps of cross-border digitisation driven by the increased awareness of the possibilities that technology can bring to the justice service provision (Kramer 2022). In all this, the attention to the technological and practical challenges of the digitisation of cross-border service of documents has been limited (Steigenga et al. 2018).

With the aim to contribute further to the body of knowledge acquired so far and go beyond aspects of theory, this study aims to examine how cross-border document service works within the EU legal landscape by looking at how it is applied by local actors, so that considerations can also be made about the role that ICT can actually play in supporting its better functioning. Analysis has been based on a mixed research method combining desk and empirical study (both qualitative and quantitative). An examination of the legal framework and an extensive literature review have been carried out in the first place, to create a picture of the application of the Service Regulation. Building on the experience developed in over ten years of research on cross-border judicial proceedings, particular attention was given to identifying gaps and divergence emerging from the interplay between EU provisions, domestic rules and national practices.

The desk research was then complemented with an empirical study aimed at understanding how the EU rules are being applied and perceived by the practitioners playing a role in the application of the procedure and its users. This included a data collection exercise carried out through two online surveys and semi-structured interviews of key informants. A first online survey was aimed at collecting information and opinions of bailiffs working at the Italian authorities competent for transmitting and receiving judicial and extrajudicial documents to be served (Bailiffs Survey). This is central to the study because it is addressed to the national agencies in charge of the cross-border transmission—both incoming and outgoing—of documents to be served. Their perspective is that of the actors of the transmission mechanism established by Arts. 4 of the Service Regulation and is therefore crucial in order to elaborate reflections on the operation of this instrument and the potential offered by the digitalisation process foreseen. A second survey addressed to lawyers (Lawyers Survey) was then conducted with the intention of complementing the results of the first one with an "external" perspective, i.e., that of the main users of UNEP

---

[17] Regulation (EC) No 1896/2006 of the European Parliament and of the Council of 12 December 2006 creating a European order for payment procedure.

[18] Regulation (EC) No 861/2007 of the European Parliament and of the Council of 11 July 2007 establishing a European Small Claims Procedure.

services, who support their clients from the preparation of the documents to be transmitted through all the subsequent steps.

Both the questionnaires were structured in five parts (46 questions in total for receiving and transmitting authorities and 40 for lawyers). In addition to demographic and profession related data, participants were asked about their daily practice and the obstacles they usually encounter within both the active and passive procedures. The lists of questions were drafted by the project team, and before being finalised it was reviewed by experts with a proven track record in their respective fields. The questionnaires were administered using Google Forms, because of the respondents' widespread familiarity with Google Suite tools and its easiness of use.

As far as bailiffs are concerned, the request to participate to the survey was sent to all 142 UNEP operating in Italy[19] and also through the National Association of Bailiffs (Associazione Ufficiali Giudiziari in Europa—AUGE).[20] The survey received 41 answers from experienced practitioners having, for the majority, more than seven years of experience in this field (80% of respondents).[21] With respect to lawyers, the request to participate was sent through the Italian National Bar Council (Consiglio Nazionale Forense CNF)[22] and received 72 answers.[23] Respondents here have a varied professional background, in terms of the size of the firms within which they work and their experience with respect to handling cases concerning cross-border cooperation in civil matters. However, the majority of professionals involved in this study declared to work in small firms (1 to 5 units in 72.9% of cases) and to have no (43.7%) or limited (21.1%) experience in the field.[24]

Taking into account the volume of potential respondents, the number of completed surveys seems to reach low numbers in absolute terms. However, in order to more accurately determine the response rate and thus make sound evaluations about the accuracy of the results, it is necessary to make some clarifications. First, it must be noted that the handling of this procedure is still considered a 'niche market' in Italy. Practitioners are not familiar with the cross-border service of documents under the EU Regulation, partly because vocational training in this field is poor, partly because of the limited number of incoming and outgoing procedures. While not relying on official data (which is not available), key stakeholders contributing to the research confirmed that within the domestic landscape the application of the Regulation (EC) 1393/2007 does not involve a large number of cases. Furthermore, it mainly involves those geographical regions with the strongest entrepreneurial vocation and a more advanced local economy that rely on internationally oriented productive activities and public entities. Beyond that, further clarification can be

---

19   The number of bailiffs in service at UNEP at the time the study was conducted was 1427. See Personnel—Transparent administration—Unep staff annual accounts Department of Judicial Organisation, Personnel and Services, Table 3—Service and protest office staff in service on 31 December 2018, last update: 4 June 2020. Available online: https://www.giustizia.it/giustizia/it/contentview.page?contentId=ART276854&previsiousPage=mg_14_7 (accessed on 20 September 2022).

20   There were about 400 Bailiffs members of the Association at the time the survey was sent out.

21   Quality of the answers was good. The survey can be divided into main questions, questions of clarification to the answer given to the previous question (11), and optional requests for comments (4). All respondents answered all main questions and answered them in full. Not all respondents answered questions that concerned requests for clarification regarding answers given to previous questions or optional requests for comment. Results for questions that did not receive a significant number of responses were not considered for the purposes of this study.

22   The Italian National Bar Council—CNF is the highest institution of the Italian legal profession. It plays a central role in the organisation of the legal profession and plays a crucial social function, in protecting legal rights, in association with the Italian Government and the judicial Order.

23   The quality of the answers is overall lower than in the Bailiffs Survey. The questionnaire is divided into main questions, questions of clarification to the answer given to the previous question (9), and optional requests for comments (3). All respondents answered almost all of the main questions, but did not always do so in full. The non-response rate was most pronounced for questions that concerned requests for clarification of replies given to previous questions or optional requests for comment. In any case, for the purposes of this study, only results concerning questions that received a significant number of responses and complete answers were considered.

24   The 87.2 percent of lawyers participating in the survey said they handle less than 5 cross-border notification procedures per year.

made by distinguishing between the two surveys. As far as the Bailiffs Survey is concerned, given the specific nature of the procedure in question and the estimate number of actors who supposedly handle European service procedures in their daily work, this sample did, actually, meet expectations, exceeding the usual threshold for the survey response rate which is between 10% and 20%.[25] Different considerations, on the other hand, must be made with respect to the Lawyers Survey. Given the very large target population,[26] the results obtained certainly cannot be subject to any generalisations. Nevertheless, they can still be useful for the purpose pursued, i.e., to complement the picture provided by the Bailiffs Survey (which constitutes the central part of the study) by offering information on the users' perspective. In addition, these numbers could be considered an indication of the lawyers' poor familiarity with Service Regulation and of the minimal number of them who specialise and actually use it.

To better understand such a transnational process, the data obtained through the surveys were then crossed with the information gathered through five additional semi-structured interviews with a selected group of experts (i.e., experienced lawyers and officers of the receiving and transmitting agencies). Interviews were conducted, in person or by phone, from September 2018 to March 2019. The interview questions were divided into four parts, for a total of 25 questions. The interviews collected information about the respondents, their role in carrying out the procedure, and how they deal with the service process in their daily practice. The questionnaires were tailored for the specific professional category of persons interviewed.[27]

The semi-structured interviews fit well with the research's explorative objective as they allowed the flexibility required to gather the needed information. The respondents had the opportunity to elaborate on specific topics, sometimes bringing forward issues not previously foreseen and that the interviewer could follow in subsequent discussions to test their extension or generality. The inputs obtained have been transcribed and then elaborated through human coding to be examined in light of the theoretical framework outlined through the desk research.[28]

### 3. Serving a Document in the EU—The Dialogue between Transmitting and Receiving Agencies "On the Books"

An effective legal framework for judicial cooperation in the field of the service of documents is a keystone for the effective functioning of the area of freedom, security and

---

[25]  Based on the interviews conducted with key informants, the UNEP offices do not have staff with expertise in the handling of Regulation 1393/2007/EC procedures. As will be further explained in Section 4, the organisation of the offices does not envisage this type of specialisation and no training courses are available on the subject, both at a local and central level. The most well equipped offices usually have just one m ember of staff to deal with these procedures when necessary. Overestimating, if 142 offices have at most one person dealing with these procedures, 28% of all potential respondents answered the questionnaire.

[26]  The number of practising lawyers registered with the Cassa Forense in the period in which the survey was conducted was 231,446. See https://www.cassaforense.it/riviste-cassa/la-previdenza-forense/previdenza/i-numeri-dell-avvocatura-2019/ (accessed on 20 September 2022).

[27]  The respondents were selected using the so-called purposive sampling method. A purposive sampling technique is a judgmental sampling technique employed when a random selection is not adequate and the knowledge of the researcher about the population to be selected is necessary to accomplish this task. This is used when a random selection of respondents cannot result in acquiring the researched information. See (Francis 2011, pp. 24–26).

[28]  In line with the Regulation (EU) 2016/679 on the protection of natural persons with regard to the processing of personal data and on the free movement of such data (GDPR) and as ethical procedures for academic research require, the respondents have been duly informed about the features and purpose of the research project, their involvement in it and how the information resulting from their contribution are used. To that purpose, an explanatory note was included at the top of the survey online and a consent form was submitted to and signed by each expert before starting the interviews. Notably, the latter document also clarified that all personal information would be subject to *pseudonymisation* to the extent possible, consistently with the needs of the study, and as early as possible in the data processing. Furthermore, participation of the interviewed in the study was solely on a voluntary basis, without incentives of any kind; the respondents have been made aware of their right to abstain from participation in the study, stop the interview or withdraw from the research at any time.

justice, as referred to in the Treaty on the EU (Mc Clean 2002; Dominelli 2018).[29] Civil and commercial court proceedings with cross-border implications often entail the need to serve documents in other Member States (e.g., when the parties live in different States, or in the case a foreign witness has to be heard). However, the transmission process is usually hampered by the incompatibility of domestic procedures and by differences in national laws as to how exactly documents should be sent, how their receipt should be confirmed, and what happens in case of failure to perform the service (Mańko 2019). It follows that both litigants can suffer additional costs and legal uncertainty.

Performing an effective service of documents is an essential part of every judicial proceeding. The good administration of justice is highly dependent on the swift and safe running of the transmission procedure, and so is the protection of parties' rights. When the document initiating proceedings is not served correctly, the rights of the defendant—as enshrined by the EU Charter of the Fundamental Rights and in the national constitutions—cannot be adequately exercised. Likewise, the legitimate expectation of the claimant to have a reliable, speedy and low-cost transmission procedure is neglected (Gascón Inchausti et al. 2017; Gascón Inchausti 2017). For these reasons, common procedures have been adopted to guarantee legal certainty and effective access to civil justice in cross-border proceedings and simplify transnational mechanisms of cooperation between civil courts.[30]

At the time of writing this article, within the EU this matter was governed by the Regulation 1393/2007 (Service Regulation) and remained so until the end of June 2022. For this reason, in the following pages (paragraphs 3 and 4) the main focus will be on this legislation and the ways in which it has been practically deployed during its time in effect. As mentioned in the introduction, by providing a better understanding of the practical issues involved in the concrete application of this method of transmission, the findings presented will allow a better prediction of the concrete changes that Regulation 2020/1784 brings about with its implementation (paragraph 5).

The Service Regulation was introduced to improve and expedite the cooperation system between the Member States in relation to the cross-border service of judicial documents in civil and commercial matters.[31] Its scope of application also covers extrajudicial documents, the service of which may be required in a variety of out-of-court proceedings, or even in the absence of any judicial proceeding.[32] The purpose of these rules is to define a simple and reliable regime for the rapid and effective execution of the transmission procedure abroad and to allow any legal person and resident in the EU to be aware of proceedings pending in any other Member State, so as to enjoy a proper defence. The Service Regulation relies on common minimum standards relating to the protection of the rights of defence,[33] and fast-track channels for forwarding documents from one Member State to another.

---

[29] This is all the more true, if one considers that a wide use of the Regulation is estimated. Data about the total number of service procedures carried out on a yearly basis are not available, however based on the fieldwork interviews carried out for the European Commission, it is estimated that as concerns the type of cases covered by the Regulation, in the timeframe 2000–2017, the number of legal proceedings in which the Regulation was applied increased from 2.8 million in 2000 to around 3.2 million in 2017 (+16%). Furthermore, it is estimated that in 2018, nearly 3.4 million civil and commercial court and out-of-court proceedings having cross-border implications required the application of the Regulation. Cft. Proposal for a Regulation of the European Parliament and of the Council amending Regulation (EC) No 1393/2007 of the European Parliament and of the Council on the service in the Member States of judicial and extrajudicial documents in civil or commercial matters (service of document) Brussels, 31 May 2018, COM(2018) 379 final, 2018/0204(COD). Please note that, as indicated in the proposal, the figures mentioned above have been estimated. See also Commission Staff working documents evaluation, accompanying the document proposal for a Regulation of the European Parliament and of the Council amending Regulation (EC) n. 1393/2007, 2018, p. 12.

[30] Within the Union, specific legislation replaces the earlier and more cumbersome system of cross-border service of judicial documents based on international conventions, such as the one provided by the Convention of 15 November 1965 on the service abroad of judicial and extrajudicial documents in civil or commercial matters and the Convention of 18 March 1970 on the taking of evidence abroad in civil or commercial matters.

[31] Regulation (EC) n. 1393/2007, art. 1.

[32] Regulation (EC) n. 1393/2007, art. 16.

[33] Regulation (EC) n. 1393/2007, art. 8.

As to the former, the Regulation aims to favour better access to justice and guarantee the procedural rights of the parties involved. Specific provisions were set to overcome linguistic barriers and ensure the understandability of the document served. A special rule providing a double-date system to determine the service date is also foreseen, aiming to strike a balance between the conflicting interests of both the applicant and the addressee of the service.[34]

As to the latter, four different methods of transmission were provided by the Regulation: (a) the "standard" procedure that relies on the direct dialogue between designated transmitting and receiving national agencies acting as intermediaries between the applicant and the addressee; (b) the direct postal service on persons residing abroad by registered letter with acknowledgement of receipt or equivalent; (c) the so-called direct service, under which any interested person can effect service of judicial documents directly through the competent authorities of the requested State;[35] (d) finally, the diplomatic and consular channel, to be used in exceptional circumstances.[36]

This paper focuses on the "standard" service process based on transmitting and receiving agencies' structure, which constitutes the Service Regulation's main trait. This method establishes a decentralised European system of documents' transmission, the cornerstone of which is the direct exchange between national authorities or bodies designated by each Member State, with territorial responsibility. Such an exchange is carried out through the collaboration between national bodies, which according to their internal procedural law are competent for the service abroad and may directly send a request to the foreign receiving authority. The transmitting agency can forward the document using "any appropriate means", in so far as the contents are true and faithful to the text forwarded and all information in it is easily readable. On the other hand, the receiving agency must perform the service—or have the document served—following the legal arrangements foreseen by the legislation of the Member State addressee. The document can also be served in line with the procedure requested by the forwarding agency, but only if the legislation of the requested Member State allows this. It is worth stressing that, within this framework, Central Authorities must be designated at the national level to play a supporting role, providing information or solving difficulties that may arise during the transmission process. Only in exceptional cases (or at the request of a transmitting agency), they are allowed to forward requests for service abroad.

This simplified cooperation system is designed to perform the service process straightforwardly and ensure the transmission process's speed, certainty and efficiency. To achieve these objectives, the procedure includes specific deadlines, and standard forms that must be used to complete every single step of the transmission process.

By and large, the service process can be divided in three main phases (Amato 2019):

(a) Start-up phase. This is carried out according to the law of the requesting State. The interested person asks the Transmitting Agency with territorial jurisdiction to send (usually by registered mail) a judicial or extrajudicial document to the Receiving Agency, which is located in the Member State where the document is to be served.

---

[34] According to article 9, generally the lex loci actum applies. Thus, the date of service is the date on which the document is served in accordance with the law of the requested Member State. This allows the addressee to rely on the domestic law of the Country where s/he lives to calculate the time period in which s/he can answer the claim. However, as an exception to this rule, if the law of the requesting Member State requires to serve a document within a particular period of time, the date to be taken into account with respect to the applicant shall be that determined by the lex processus. This will also make possible to protect the claimant, in the case that s/he has an interest in acting within a given period or a specific date, avoiding that events which fall outside his/her own control could affect the positive outcome of the service procedure.

[35] Based on the Service Regulation, direct service can be effected only in the case that this is permitted under the law of that Member State.

[36] The first one relies to transmission by consular or diplomatic channels. This is an indirect forwarding mechanism, under which the service is carried out by the Consul of the requesting State on the addressee. Service can also be performed by diplomatic or consular agents directly, in the case that service is effectuated by the Consul of the requesting State on the addressee.

(b). Forwarding phase. The document is transmitted according to the Service Regulation requirements. The document is complemented with a standard multilingual form provided by the Regulation, filled out in one of the languages that the Member State of destination accepts. The form includes the official request for cooperation, together with the details of both the applicant and the addressee and the specifics concerning the method of transmission to be used. The form also provides the details of the type of act that is forwarded (e.g., the language in which this was written, and the translation possibly attached).[37]

(c). Delivery phase. This is carried out in the requested State and is, thus, mainly covered by the relevant domestic legislation. Once the document has been delivered to the Receiving Agency, an acknowledgement of receipt must be sent back to the Transmitting Agency within seven days. In case of improper service (e.g., if the document is out of the scope of the Regulation or in case of failure to comply with the compulsory formalities required) the document must be returned to the sender. It is worth highlighting that under the EU regime, dialogue between national agencies for consultation purposes is encouraged; for this reason, when gaps in the information needed risk to hamper the proper service of the act forwarded, the Receiving Agency should contact the foreign authority to acquire the missing elements and successfully conclude the service process. Only in the event that service cannot be made within one month (e.g., because the addressee cannot be located), the document can be returned together with a certificate of non-service (Amato 2019).

This streamlined system represents a breakthrough compared to the traditional legal regime. It is designed to remove obstacles usually resulting from the differences between judicial and administrative systems and guarantee the same speed and reliability when the service is performed within the domestic jurisdiction. Previous survey-based analysis showed that stakeholders perceive these EU rules as having improved the cross-border service process, positively impacting the functioning of the area of freedom, security and justice.[38] Despite this, the policy goals pursued by the Regulation are still hindered by gaps and ambiguities in the legislation and limits deriving from its practical implementation. Notably, the method of transmission based on the intermediation between receiving and transmitting agencies has turned out to work slower and less efficiently than expected, with delays encountered in each phase of the process. Suffice it to say that the average number of days required for having the document served is significantly above the deadline set out in the Regulation (amounting to just one month) and that this result is particularly worrying for proceedings in which timing is crucial.[39]

---

[37] With regard to this, under the Regulation, the Transmitting Agency has the duty to advise the applicant that the translation of documents to be forwarded may be needed and that the addressee has in principle the right to refuse them in the case that this requirement is not met or if the document is drawn up in a language that s/he is not able to understand.

[38] Cft. MainStrat, Study on the application of Council Regulation (EC) No 1393/2007 on the service of judicial and extra judicial documents in civil or commercial matters, report realised upon request of the EU Commission, 2014, ISBN: 978-92-79-34791-7, pp. 52–53.

[39] The 2013 evaluation carried out by the Commission estimated that the average days required for having a document served was about 3.3 months, significantly above the one-month deadline set out in the Regulation. This is highly worrying if taking into account that timing is crucial, especially for the proper carrying out of certain types of proceedings (e.g., in insolvency or enforcement cases). In the most recent assessments available, a precise estimation of the time taken to complete the request is lacking; however, it is clearly stressed that documents are not served within one month of the request being received by the receiving agency. See the Report from the Commission to the European Parliament, the Council and the European Economic and Social Committee on the application of Regulation (EC) No 1393/2007 of the European Parliament and of the Council on the service in the Member States of judicial and extrajudicial documents in civil or commercial matters (service of documents) COM/2013/0858 final, pp. 9 and 17, and the Commission Staff Working Document Impact Assessment Accompanying the document Proposal for a Regulation of the European Parliament and of the Council amending Regulation (EC) No 1393/2007 of the European Parliament and of the Council on the service in the Member States of judicial and extrajudicial documents in civil or commercial matters (service of documents) SWD(2018) 287 final, p. 19 and Annex VIII, p. 27; Report from the Commission to the European Parliament, the Council and the European Economic and Social Committee on the application of Regulation (EC) No 1393/2007 of the European Parliament and of the Council on the service in the Member States of judicial and extrajudicial documents in civil or commercial matters (service of documents) COM/2013/0858 final.

The underlying problem is the limited competence the EU can exercise in this matter, which makes it difficult to establish a common European notion of service of documents. The Service Regulation is based on the principle of national procedural autonomy,[40] according to which the EU rules can only regulate the transmission and service of documents in transnational relations, but they cannot intrude in the domestic procedural laws. This creates a mismatch between the European and national dimensions, where a standardisation process is hardly accepted and difficult to realise from a political and operational viewpoint.[41]

In this respect, the differences in the national procedures are only part of the problem.[42] Recent studies have shown that much depends on how national actors interpret and apply the relevant provisions within the domestic domain. Supranational provisions' wording includes ambiguities, thus favouring misinterpretations and non-uniform interpretations of the same rule.[43] An example is provided by the language requirements of the document to be served. These requirements are meant to protect both the legitimate interest of the recipient—who can refuse the service of a document in a language that s/he cannot understand—and those of the claimant, because such a refusal does not make the service invalid, but it is considered as a 'mistake' that can be fixed forwarding the documentation in the correct language. Despite the good intentions, such a solution inevitably results in further delays and unwanted practical consequences in the absence of guidance on a variety of issues, such as: how to evaluate the actual ability of the addressee to understand the language; on which standard to rely to conduct the evaluation (Bohunova 2008; Galic 2013)[44] who is in charge to conduct such an assessment; which degree of knowledge should be required to legally refuse the document served, etc.

Also, the lack of familiarity of the actors involved in carrying out the procedure with the European legislation plays a key role. For example, it seems that the collaboration between agencies is often obstructed by systematic deviations from the relevant supranational rules. The tendency to not properly apply the so-called 're-transmission duty' clause, as foreseen by art. 6 (2) of the Service Regulation, exemplifies this situation well.[45] The analysis of domestic working practices has revealed that when the document to be served is forwarded to the wrong receiving agency, this is usually returned to the sender, and no effort is made to identify the territorially competent authority. Action to favour the finalisation of the procedure, assisting the transmitting agency in identifying the competent local authorities, locating the addressee or clarifying the address on the document,[46] is taken only if the receiving agency has experience with the procedure. The same can be said with regard to the incorrect use of the standard forms. These forms are one of the pillars of the EU service process; despite this, they are often filled in incorrectly, miss

---

[40] In accordance with the principle of subsidiarity and with the provisions laid down in Article 81 of the Treaty of Lisbon.

[41] See Working Party on e-Law (e-Justice), Draft report of the expert group on the e-Service of Documents, Brussels, 19 July 2018, 11275/18.

[42] DMI in Consortium with University of Florence and University of Uppsala, Study on the service of documents. Comparative legal analysis of the relevant laws and practices of the Member States. Final Report (No JUST/2014/JCOO/PR/CIVI/0049), 5th October 2016.

[43] See Commission Staff Working Document Impact Assessment Accompanying the document Proposal for a Regulation of the European Parliament and of the Council amending Regulation (EC) No 1393/2007 of the European Parliament and of the Council on the service in the Member States of judicial and extrajudicial documents in civil or commercial matters (service of documents), SWD/2018/287 final.

[44] See Case C-14/07, Weiss, ECLI:EU:C:2008:264.

[45] Commission Staff working documents evaluation, accompanying the document proposal for a Regulation of the European Parliament and of the Council amending Regulation (EC) n. 1373/2007 of the European Parliament and of the Council of the 13 November 2007 on the service in the Member States of judicial and extra-judicial documents in civil and commercial matters (service of documents), 2018, p. 43.

[46] Commission Staff working documents evaluation, accompanying the document proposal for a Regulation of the European Parliament and of the Council amending Regulation (EC) n. 1373/2007 of the European Parliament and of the Council of the 13 November 2007 on the service in the Member States of judicial and extra-judicial documents in civil and commercial matters (service of documents), 2018, p. 31.

important information or are not readable.[47] In addition, administrative formalities, the heavy reliance on paper-based means of communication,[48] and various language-related problems further hamper the cross-border dialogue between national actors.

### 4. The Service of Documents 'in Action': The Italian Experience

As seen above, the Service Regulation establishes a decentralised regional system of documents' transmission, which relies mostly on the direct collaboration between national authorities or bodies designated by each Member State to act as transmitting and receiving agencies. This format is designed to speed up and make the entire process easier, by setting up specific deadlines and providing standard forms that are also intended to improve the certainty and efficiency of the transmission procedure.

However, this transmission channel has proven to be underperforming, even considering the potential of using technology.[49] This is the result of a varied set of factors, which include: legal barriers, lack of interoperability between national systems, and old entrenched habits.[50] Also, as with other European civil cooperation procedures, the practical application of the Service Regulation is characterised by legislative gaps that need to be filled by constant reference to national procedural rules, which means that its implementation requires close coordination between the supranational and national legal dimensions. This exercise, however, which is challenging *per se*, becomes even more difficult due to the multitude of local practices that deviate from the letter and spirit of the regulation, thus hindering the proper application of the procedure (Amato 2019).

In order to identify those issues—also in view of the process of digitalisation of this procedure—the following paragraphs will present the results of the empirical research conducted in Italy. This aims to provide an in-depth view of the application of the European rules at the domestic level, offering a clearer picture of how the Service Regulation works in practice and how the main actors deal with this cross-border procedure in their daily practice.

### *4.1. A Look at the Main Actors: The Receiving and Transmitting Agencies*

According to the Service Regulation, the national body that is competent under domestic procedural law to serve documents abroad may send a request directly to a foreign receiving agency. Likewise, this authority can directly receive the documents to be notified from another Member State.

In Italy, the agencies competent for receiving and transmitting documents to be served are the *Uffici Notificazione Esecuzioni e Protesti* (Notification, Enforcement and Protest Office), better known as UNEP.[51] The Notification, Enforcement and Protest Offices operate within the Italian Courts of First Instance and Appeal, but have organisational, administrative and budgetary autonomy. They are specialised offices responsible for serving acts and executing orders at the request of private parties and judicial authorities in civil and criminal matters. Notably, in the context of civil proceedings they must ensure the service of judicial and extrajudicial documents within the domestic and international jurisdictions, thus including the service of documents to be notified abroad under the Service Regulation.

---

47   MainStrat, Study on the application of Council Regulation (EC) No 1393/2007 on the service of judicial and extra judicial documents in civil or commercial matters, report realised upon request of the EU Commission, 2014, ISBN: 978-92-79-34791-7.

48   See note 45 above.

49   MainStrat, Study on the application of Council Regulation (EC) No 1393/2007 on the service of judicial and extra judicial documents in civil or commercial matters, report realised upon request of the EU Commission, 2014, ISBN: 978-92-79-34791-7.

50   See Proposal for a Regulation of the European Parliament and of the Council amending Regulation (EC) No 1393/2007 of the European Parliament and of the Council on the service in the Member States of judicial and extrajudicial documents in civil or commercial matters (service of documents), 2018/0204(COD).

51   R.D. 30 January 1941, n. 12 (Judicial System), Article 3 and following modifications.

All UNEP have been designated Transmission Agencies for Italy and are therefore responsible for forwarding the documents to be served abroad.[52] They have a widespread presence throughout the national territory. According to the official Ministry of Justice data, there are currently 142 UNEP in operation (113 UNEP at the Courts of First Instance and 29 at the Courts of Appeal),[53] within which an average of 10 staff members work.[54] The staff—recruited through an open competition held by the Ministry of Justice—is divided into three job profiles: (a) judicial officials (*funzionario giudiziario*), (b) bailiffs (*ufficiale giudiziario*), and (c) judicial operators/assistants.[55] As Transmitting Agency, the territorially competent UNEP must first check whether the forwarded document falls within the Regulation's scope. Accordingly, the office has to verify the civil or commercial nature of the document to be forwarded and its formal accuracy. It also has to inform the applicant (or his/her attorney) of the possibility of the addressee rejecting the document if a translation is not provided in one of the languages indicated explicitly in the Regulation.[56] The UNEP staff then has to make sure that the accompanying standard form is duly filled in (or assist the party or his/her attorney in completing) and attached to the document for which service is requested.[57]

With regard to the receipt from abroad of documents to be served in Italy, the approach adopted at the national level departs widely from what is established by the Regulation as it provides for centralised jurisdiction. In Italy, in fact, the only competent authority under the passive service procedure is the UNEP at the Court of Appeal in Rome, which, at the same time, is also the designated central authority.[58] In essence, UNEP at the Court of Appeals in Rome is not limited to ensuring the tasks of assistance and support in the transmission phase, but is also the only authority in charge of receiving requests for notification from other member states and redirecting them to the territorially competent offices. This is noteworthy considering that the number of purely national notification procedures alone handled each year is approximately 9000 and that, at the time the interviews were conducted, only four full-time staff members within UNEP at the Court of Appeal in Rome were assigned to notification activities.[59]

Overall, the findings show that the organisational structure (including the human and instrumental resources available to all the UNEP) appears ill-equipped to deal effectively with the European service system designed by the Service Regulation. On the one hand, UNEP staff seem to be lacking in some so-called 'soft' skills, which are nevertheless essential for the efficient and effective functioning of the procedure; on the other hand, they are poorly versed in the relevant European legislation it is called upon to apply.

As an example, Bailiffs Survey gives an account of a low to medium IT skilled staff, believed to be able to use the basic equipment typically made available in public administrations, such as computers, scanners, fax machines, internet connection, and institutional email addresses (created and made available to UNEP staff only in 2014). At the same time, the situation seems to be more challenging in terms of language skills. Based on the data

---

[52]  Article 2 Service Regulation.

[53]  See "Giustizia Map" at the official Ministry of Justice website https://www.giustizia.it/giustizia/it/mg_4.page (accessed on 25 July 2022).

[54]  See Personnel—Transparent administration—Unep staff annual accounts Department of Judicial Organisation, Personnel and Services, Table 3—Service and protest office staff in service on 31 December 2018, last update: 4 June 2020. Available online: https://www.giustizia.it/giustizia/it/contentview.page?contentId=ART276854&previsiousPage=mg_14_7 (accessed on 20 September 2022).

[55]  For example, secondary school for the profile of judicial operator, second level master's degree in law, economics, or political science for the profile of judicial official. On the basis of the interviews conducted, the last open competition held to recruit UNEP staff has been carried out in the 1990s.

[56]  Article 8 Service Regulation.

[57]  Annex I of the Service Regulation.

[58]  Article 3 Service Regulation.

[59]  With regard to this, it is worth noticing that the file management systems (Gestione Servizi UNEP—GSU web) in use at each Office is not able to include procedures applying the Service Regulation. The latter are thus managed through separate electronic registration systems (in the case of the UNEP at the Court of Appeal in Rome an excel file).

collected, UNEP staff are not sufficiently equipped to properly carry out tasks involving constant contact with foreign jurisdictions and actors. Foreign language speakers seem to be a limited group. Among the respondents, only 10% declared to have an excellent knowledge of at least one foreign language (French being the most commonly known).[60] This may be primarily due to the seniority of UNEP staff (most of whom are nearing retirement) and the lack of attention to organising re-skilling and training courses over time.

Furthermore, with the exception of UNEP at the Court of Appeal in Rome, staff are generally not experienced in the service of cross-border documents. Typically, within each office, there are no specialised units or focal persons in charge of international activities. Only a few offices can rely on expert officials to apply the service regulations and communicate with foreign authorities when necessary. Notably, based on the results of the Bailiffs Survey, only 5% of the respondents declared to have an 'excellent' knowledge of the Service Regulation. Such expertise has been acquired through professional courses undertaken on a voluntary basis (outside the professional context) or thanks to the experience gained in the field. The professionals interviewed confirmed this picture, declaring that no vocational training is ever organised at the central or local level. Likewise, official guidelines on the practical application of the Regulation have never been drawn up by the Ministry of Justice. At present, the only official instructions available are those published by the Ministry of Foreign Affairs for the notification abroad of judicial and extrajudicial acts in civil and commercial matters (2019 Edition), which provides only some general directions, but fails to give any of the operational information needed by practitioners to support their daily practice.[61] Guidance is sometimes offered through working tools or information sheets drafted or adopted at a local level and made available online on the websites of the single Courts. These can undoubtedly be a source of helpful information for UNEP staff and the public, (e.g., on how to fill out standard forms and the operational steps to follow in order to perform the service procedure); nevertheless, since they do not adhere to a common approach, they are likely to encourage the adoption of different practices within the national context. The use of different operational practices is evident from the Lawyers Survey, the results of which give an account, for example, of different ways of submitting the required documentation or differences in the way in which service fees must be paid. In this respect, it is worth noting that the approach taken is also uneven with respect to the translation of documents for which notification is required. In particular, lawyers point out that although the Regulation does not place the notarisation of the translation as mandatory, in some UNEPs this is considered a requirement to be met in order to go ahead with the procedure.

For direction on practical aspects related to the application of the Regulation (e.g., availability of forms, clarification on the interpretation of provisions), respondents said they refer to the EU justice portal to seek information (48%) or they turn to colleagues considered more experienced (41%). A small but significant percentage do not know where to turn (15%) and sometimes conduct general Internet searches to find what they believe to be the information they need.

In this regard, it is worth mentioning the role played by professional networks of judicial officers working nationally and internationally, which seem to counterbalance the lack of a centrally provided guide. One such organisation is the Italian Association of Judicial Officers in Europe,[62] whose website includes a special section that provides operational information on the application of the Service Regulation, standard forms to be used to perform each step of the procedure, as well as an e-learning platform for practitioners. This national association is part of international umbrella networks that

---

[60] According to the results obtained through the survey online.

[61] See: the "Guida alla notifica all'estero di atti giudiziari ed extra giudiziali in materia civile e commerciale (Edizione 2019)". Available online: https://www.esteri.it/mae/resource/doc/2019/02/guida_notifiche_civile_2019.pdf (accessed on 30 June 2020).

[62] The association was created in 2005 to organise training; to give advice to members on problems arising in the daily practice; and to draw up proposals aimed at strengthening the efficiency of the justice service and the consequent national economic development, by promoting the role of the bailiffs.

brings together the bodies representing bailiffs in several Member States. At the European level, for example, the Association is a member of the European Chamber of Judicial Officers (CEHJ)[63] that drafts position papers on European regulations, both as they are being drafted and as they are being reformed, in order to bring the experience of judicial officers to European forums, along with best practices and recommendations for improving the use of available cooperation tools.[64]

*4.2. A Look at the Procedure 'in Action': Receiving and Transmitting a Document in Italy*

4.2.1. The Active Procedure in Operation: What Really Happens during Start-Up and Forwarding Phases

Starting the process of transmitting a document to be served abroad under the Regulation is a seemingly straightforward operation. The relevant EU rules set indeed a transmission mechanism that is intended to be simple, user-friendly, and under which standard forms must be used precisely to guide the user and minimise any leeway for error. This is why the sender, who is responsible for preparing all the paperwork required for transmission, is entitled to initiate and follow the procedure autonomously, with no need to be supported by a lawyer.

However, this is unlikely to be the case in practice. In the experience of the interviewees, in the Italian territory, the tasks related to the drafting of standard forms, the overall preparation of the paperwork and its submission to the territorially competent UNEP, are always carried out by lawyers.[65] There is a general consensus among participants that this procedure is beyond the reach of an average user. Thus, the support of a professional, whether a lawyer or a specialised official, is deemed necessary.[66] However, based on the results of both surveys, it cannot be overlooked that this kind of professional support may not be widely available. On the one hand, the percentage of UNEP staff replying to the questionnaire who declared to have expertise in this matter is extremely low. Likewise, most professional users seem unfamiliar with the European notification procedure and, more generally, with the salient issues of conducting a notification abroad. The percentage of lawyers who say they have no training of European cross-border cooperation procedures in civil matters is significant (45.1%) and even higher is that of those who say they conduct less than five cross-border notification procedures per year (87.2%).

Needless to say, this has some major consequences when requesting the initiation of service proceedings, where it is crucial to fill out the forms correctly, prepare the necessary translations of the documents to be served, and identify the precise address of the person to be served. In everyday practice, when a document must be notified abroad under the Service Regulation, the party's lawyer submits it in paper form to the UNEP territorially competent for its transmission abroad along with the attached forms. Here, front-office clerks only conduct some routine checks on aspects that, although quite basic, cannot be taken for granted. For example, they check that the document to be served is within the scope of the Regulations, whether the standard form has actually been attached,[67] is complete, bears the address of the addressee, and has been drafted in a language version

---

[63] European Chamber of the Judicial Officers ("CEHJ") official website http://www.cehj.eu. The Chamber operates since 2012 to ensure the sustainability and development of the work undertaken by consortia of judicial officers and to take an active part in the EU legislative making process promoting bailiffs' interests. At present, the EUBF (2018) continues the activities and project of the CEHJ (e.g., "Comparative study on the application of Brussels I bis", "Find a Bailiff 2, in which the Chamber acts a project leader, and "European Judicial Training for Court Staff and Bailiffs 1 and 2, "Court Database 2 and "Me-CODEX I" it acts as a partner).

[64] The members actively worked on the reform of the EU regulation 1393/2007 on the service of documents in Europe and other measures dealing with the European cooperation in civil and commercial matters.

[65] This was declared by the 100% of the respondents to the Bailiffs Survey.

[66] This opinion was shared by the 82.5% of respondents to the Bailiffs Survey and the 91.2% of respondents to the Lawyers Survey.

[67] 17.2% of respondents declared that they do not always use the standard forms attached to the Service Regulation (Lawyers Survey).

accepted by the receiving country.[68] UNEP staff also handle the identification of the receiving authority, using the European Judicial Atlas in Civil Matters.[69] As far as the language of the document is concerned, on the other hand, this is not usually assessed, as compliance to linguistic requirements is considered by UNEP staff to be a duty on the plaintiff, not least because the defendant can challenge them.

Beyond these fundamental checking operations, no forms of user support are usually provided. Only in a few cases, UNEP front-office staff help users fill out forms,[70] provide information on how to carry out the procedure, and also make available fact sheets providing operational information and other working tools (e.g., draft standard documents for requesting UNEP's services, forms for delegating UNEP officials to complete the necessary standard forms on behalf of the party requesting their services, etc.). Nevertheless, this service is not widely and evenly available in Italy, as it is the result of local arrangements that can vary greatly from office to office. In most cases, for guidance on practical aspects relating to the application of the regulation, lawyers said they refer to the EU justice portal (40%) or rely on their personal contacts with more experienced colleagues or contact persons at the local level (64.6%). While quite limited, the percentage of those who do not know where to turn (14.3%) does remain significant.

Once the necessary checks have been completed, the document and the appropriate form are sent by the UNEP official to the foreign authority by registered mail. The Receiving Agency should reply to the Transmitting Agency as soon as possible—and in any case within seven days of receipt—by sending an 'Acknowledgement of Receipt Form'. However, based on the responses collected, this requirement is not met on a regular basis (87.5%), and the case is the same for the use of other standard forms that the Receiving Agency should forward to the Transmitting Agency at different stages of the procedure (e.g., successful delivery, service carried out, recipient not found, unknown address, or refusal of the document).[71] Lawyers, in particular, report that in the case of refusal by the recipient because the document received is not drafted/accompanied by a translation in a language he/she understands, the appropriate certificate is not regularly received by the sender (68.6%) and, even where returned back correctly, it does not always indicate the language the recipient claims to understand (40.9%). Based on the interviews carried out, it appears that in some member States receiving authorities either do not send the required forms at all—including the one certifying that notification has been made—or, even when they do fulfil this task, there are long delays due in part to the use of the postal channel. More generally, according to the experts interviewed and participating in the surveys, the time required for postal communication seriously hampers the procedure's usefulness.

4.2.2. The Passive Procedure in Operation: What Really Happens during the Delivery Phase

Concerning the passive procedure, the results of both the surveys and the interviews account for several departures from the letter and spirit of the Regulation. These departures affect—even markedly—the timeframe for completion of the procedure as its overall effectiveness and consistency. Such deviations concern both organisational choices made by Italy when transposing the relevant European legislation and the ways in which professionals apply the Regulation in their daily practice.

Looking at those deviations affecting organisational choices, the Italian decision to centralise the competence to receive notification requests from other EU member states have to be considered. As mentioned above, in Italy the Central Authority has been given

---

68　In this regard, it is worth noting that only 43.9% of the responding lawyers stated that they prepare the form in the language spoken or otherwise accepted in the receiving member state. Forty percent of respondents submit the forms in Italian.

69　See the Justice Portal at the page https://e-justice.europa.eu/content_serving_documents-373-it-en.do?member=1 (accessed on 30 June 2020).

70　13.6 percent of respondents said UNEP staff provide support in filling out forms (Lawyers Survey).

71　Only 12% of the respondents declared having received the standard forms attached to the regulation from the foreign authority.

an additional 'entry point' role and is, accordingly, in charge of sorting the incoming documents and routing them to the territorially competent UNEPs. The objective, supposedly, is to reduce the possibility of error in the selection of the competent body by the foreign transmitting authority and to avoid mistakes that could lengthen the time needed to complete the procedure. However, this solution does not seem to achieve this goal, but in a high percentage of cases, it appears to be causing a dilation of the notification time. Interviews revealed that in about 50% of cases, the office receiving the document by the central authority considers that it is not competent to perform the service and when this happens, it uses to return the paperwork to Rome for re-transmission rather than proceed to *ex officio* referral, as would be required by the Regulation.[72]

As for how practitioners enforce European legislation in their day-to-day activities, two major problems have been found that frequently impair the proper and swift conduct of the service process. The first of these stems from the reluctance of domestic agencies not to seek assistance from their counterparts abroad, as would be required by the Regulation.[73] The findings showed that errors, omissions, or inaccuracies in the documentation to be served are often a source of practical problems and a cause of delays; in particular, survey responses indicate a lack of information in the documentation received or the provision of an incorrect recipient address among the circumstances that most frequently can lead to failure of the notification procedure or prolonged delays that go beyond the maximum 30 days provided. The Regulation stipulates indeed that the receiving agency must make an effort to contact its foreign counterpart by the most expeditious means possible to obtain the missing elements and complete the service process.[74] This is because the overall objective of the Regulation is to remove obstacles arising from differences between the legal and administrative systems of EU Member States, so that citizens are guaranteed the same speed of service that they would have in proceedings conducted entirely within national borders. However, in practice, this dialogue very rarely takes place[75] or, when it does, communication by mail is used, which does not allow for a quick and effective exchange of information. In this regard, though, it must be said that such a problem does not concern only Italy. In most Member States, authorities are reluctant to consult the transmitting agency directly, partly for reasons related to the use of different languages. Also, a heavy and widespread reliance on paper media persists throughout the EU, even if some Member States accept interaction through electronic means, and national laws often provide for the possibility of using electronic channels.[76] Based on information gathered by the European Commission, the postal service remains the primary means of communication between transmitting and receiving agencies used by member States, even for correspondence intended only to obtain clarification (e.g., additional address information).[77]

Along with the hindrances resulting from the lack of effective cross-border dialogue among the agencies involved in the procedure, the second main challenge brought out by this study is connected with the language of the documents to be served. When UNEP acts as the receiving agency, it performs minimal checks on the documents to be served,

---

[72] The survey replies indicate that documentation from abroad is delivered in 97.5 percent of cases in paper format via postal service (registered mail). Only a very small percentage of requests arrive by fax. In this case, however, the request is generally refused, because very often the document to be served is not readable due to the poor clarity of faxed documents; in addition, there is also the fear that the pages do not match the original document, especially when they are written in a foreign language that UNEP staff cannot understand.

[73] Articles 6(2) Service Regulation.

[74] See note 72.

[75] Eighty-five percent of respondents to the online questionnaire said that they never followed the instruction in the Regulations to consult with the foreign counterpart to obtain missing information or to clarify certain details they deemed necessary (Bailiffs Survey).

[76] It seems that the method of working with paper documents is in line with the regular working practice of the national authorities and, thus perceived as most practical for them. Cft. Commission Staff Working Document Impact Assessment Accompanying the document Proposal for a Regulation of the European Parliament and of the Council amending Regulation (EC) No 1393/2007 of the European Parliament and of the Council on the service in the Member States of judicial and extrajudicial documents in civil or commercial matters (service of documents), SWD/2018/287 final, p. 15 and Attachment n. 8.

[77] See note 45 above.

merely verifying cases of improper service. The language of the document and the attached forms, even when potentially incorrect, does not represent an obstacle to the completion of the service procedure. 79.5% of the judicial officers who participated in the survey said that the paperwork is generally accepted when the attached standard forms are in one of the official languages of the EU, even though, according to Article 4 of the Service Regulation, the document received must be accompanied by a standard form written in one of the official languages of the receiving member State, or in a language that the receiving member State has agreed to accept.[78] As already mentioned, UNEP staff considers that it is the recipients who are responsible for this 'language check', consequently, they place the burden of accepting or rejecting the service on them.

However, this task, which is highly sensitive as it is related to the exercise of the recipients' right of defence, according to the letter of the Regulation[79] and the case law of the CJEU it entails precise responsibilities for the receiving agency. It follows that UNEP staff have important information duties that are ancillary to the genuine exercise of these guarantees and are, thus, crucial for effecting service. Notably, when effecting the service the receiving agency should inform the addressee, using the standard form set out in Annex II, which must be used mandatorily for service to be considered properly performed.[80] However, based on interviews and online surveys, this requirement is rarely met in Italy. In particular, according to the lawyers involved in the study, the addressee is often not adequately made aware of his or her rights. Furthermore, it sometimes happens that the form is not attached to the document served and, even when duly provided, its contents are generally not genuinely explained to the addressee.

This practice, in any case, is not an isolated situation and does not occur only in Italy. As shown in the 2018 Impact Assessment conducted by the EU Commission, this is actually a 'European trend', which stems from the fact that Regulation 1393/2007 is flawed when it comes to addressing language aspects, especially the obligation to provide information on the right to refuse service.[81] The CJEU has actually filled in some of these gaps with case law, stating on several occasions that the receiving agency is always obliged-without a margin of discretion-to inform recipients of their rights, systematically using the specific form provided in the Regulation.[82] Nevertheless, practice shows that information about the right of refusal is still not always provided appropriately, or even at all, because there remains a widespread notion that the relevant form is not necessary when the document to be served is already written or translated into a language understandable by the addressee or into an official language of the receiving State.[83] This has significant consequences for the recipient,

---

[78]   Bailiffs Survey.

[79]   Article 8, Service Regulation.

[80]   With respect to the information to be provided to the addressees, including of their right to refuse receipt see: CJEU, 06 September 2018, Catlin Europe SE against O.K. Trans Praha spol. s r.o., C21/17; CJEU, 2 March 2017, Case C-354/15, Andrew Marcus Henderson v. Novo Banco SA; CJEU, 28 April 2016, Alta Realitat SL against Erlock Film ApS and Ulrich Thomsen, C-384/14; CJEU, 16 September 2015, Case C-519/13, Alpha Bank Cyprus v. Dau Si Senh.

[81]   See Commission Staff Working Document Impact Assessment Accompanying the document Proposal for a Regulation of the European Parliament and of the Council amending Regulation (EC) No 1393/2007 of the European Parliament and of the Council on the service in the Member States of judicial and extrajudicial documents in civil or commercial matters (service of documents), SWD/2018/287 final, para 2.2.1.3. See also para 5.1.2.2. of the Annex 8: Evaluation Report.

[82]   Case C-519/13, Alpha Bank Cyprus, ECLI:EU:C:2015:603.

[83]   On the basis of the cases available on the Unalex database, it was often an issue before the court that the information on the right of refusal according to Article 8 of the Regulation had been misleading or not given. Cft. Commission Staff working documents evaluation, accompanying the document Proposal for a Regulation of the European Parliament and of the Council amending Regulation (EC) n. 1373/2007 of the European Parliament and of the Council of the 13 November 2007 on the service in the Member States of judicial and extra-judicial documents in civil and commercial matters (service of documents), 2018, p. 37. See also: See Case C-14/07, Weiss, ECLI:EU:C:2008:264. Cft. P.Bohunova, 'Regulation on Service of Documents: Translation of Documents Instituting Proceedings Served Abroad', sub 4, available at http://www.muni.cz/research/publications/818211; and Galic, A. 2013. *Service abroad in civil and commercial matters—from The Hague Conventions to the EU 1393/2007 Regulation. Collection of Papers, Faculty of Law, Nis* 65: 64.

who may not exercise the right to object to the receipt of the document. The sender can also be harmed because the burden to prove that the addressee understands the language in which the document was drafted remains on him or her. This assessment, by the way, is far from simple, since it cannot be merely on formal grounds. As the EU Commission pointed out, the assessment of the addressee's ability to genuinely understand the language of the document received is not a simple task and requires a thorough evaluation of several subjective factors and cannot be carried out solely on the basis of objective circumstances, such as those related to citizenship, residence or domicile, etc., which should be considered as mere indications. However, the Regulation does not provide guidance on how to conduct such an assessment and what standard should be relied upon to perform this task.

### 5. The Reform of the Service Regulation and the Digitalisation of the Service Process

The European legal regime on cross-border notifications has recently been recast with the adoption of Regulation 2020/1784 (entered into force on 1 July 2022).[84] This new Regulation was adopted with the overall objective of aligning the framework of judicial cooperation with the Digital Single Market strategy and, more precisely, to reduce the cost burden and unjustified delays for citizens and businesses involved in cross-border proceedings through faster transmission mechanisms that are less dependent on paper-based communication. For this reason, apart from some minor changes concerning the addressee's refusal to accept the document (Article 12), this reform focuses almost entirely on the use of technology and the dematerialisation of documents to be served.

This strong emphasis on the digitalisation of procedures is not an isolated initiative. It is part of a broader regulatory trend that aims to modernise judicial cooperation procedures without changing the proceedings themselves, exploiting the potential of existing ICT tools for document transmission.[85] Digitalisation is increasingly becoming an essential asset for improving the efficiency and resilience of communication channels inherent in cooperation between national authorities in EU cross-border cases and, ultimately, ensuring access to justice (Ontanu 2022). Over the years, the many legal instruments adopted to improve judicial cooperation in civil, commercial and criminal matters have conceived the work primarily as paper-based. This approach has failed to meet the speed and simplification goals these instruments were supposed to achieve. Also, the pandemic contributed to an increased sense of urgency for creating a framework that can ensure access to justice, facilitate cross-border dialogue and increase communication security. In this situation, regulatory action is needed to bridge this gap and make using technology not just an option but the preferred means of carrying out cross-border cooperation within a common legal framework providing a shared approach and compatible technical solutions (Ontanu 2022).

In line with this general approach, the 2020 Service Regulation's goal has been to improve the inter-agency system of cooperation currently in place, the workflow of which was found to be underperforming and less efficient than expected. Despite the old Service Regulation being opened up to technology,[86] in many States—including Italy—the competent authorities do not use ICT to perform cross-border notification, partly because postal service has often been designated as the sole means of receipt of documents,[87] partly because of the reluctance of the actors involved to exploit the potential of recent

---

84    Articles 5, 8 and 10 of the Regulation 2020/1784 take exception to this date of entry into force. These measures will take effect in 2025.

85    We refer to the adoption of the: Proposal for a Regulation on the digitalisation of judicial cooperation and access to justice in cross-border civil, commercial and criminal matters, and amending certain acts in the field of judicial cooperation, Brussels, 1 December 2021 COM(2021) 759 final; Regulation (EU) 2022/850 of 30 May 2022 on a computerised system for the cross-border electronic exchange of data in the area of judicial cooperation in civil and criminal matters (e-CODEX system), and amending Regulation (EU) 2018/1726; and, two recast regulations from November 2020, that is, the Regulation 2020/1784 of 25 November 2020 on the service in the Member States of judicial and extrajudicial documents in civil or commercial matters (service of documents) (recast) and the Service of Documents Regulation, which is discussed in this paragraph.

86    Article 4 (2), Regulation 1393/2007/EU.

87    See Justice Portal page about the use of the Service Regulation in Italy. Available online: https://e-justice.europa.eu/373/EN/serving_documents?ITALY&member=1 (accessed on 3 November 2022).

technological developments.[88] In Italy, for instance, even within the 'domestic segment' of the procedure—whether passive or active—only paper documents are normally accepted. The electronic copies received via PEC (certified electronic mail service, valid for service of documents in Italy) are usually refused when the request is for service of documents to be made under the Service Regulation.

Under the new rules, the mechanism of transmission between national agencies is not altered in its essential features. If anything, there is a strengthening of the role of these agencies, which are placed as key intermediaries between the Member States to best deliver a document, to preserve not only the rights of the sender but also, and more importantly, the rights of the recipient. The only aspect affected by the new Regulation is the medium through which such a transmission must be performed. Indeed, the exchange of paper-based documents and communications is entirely replaced by the mandatory use of a secure and reliable decentralised system for data exchanges that builds upon a network of national ICT systems and interoperable communication infrastructure based on nationally deployed access points, operating under the individual responsibility and management of each Member State.[89]

The system ultimately chosen for information exchange is the e-CODEX infrastructure (Velicogna 2014; Carboni and Velicogna 2011; Velicogna 2015; Velicogna and Lupo 2017).[90] e-CODEX (e-Justice Communication via On-line Data Exchange) is a tool specifically designed to facilitate the cross-border electronic exchange of data in the area of judicial cooperation in civil and criminal matters, the aim of which is to improve the efficiency of cross-border communication between competent authorities and to facilitate citizens' and businesses' access to justice. Based on the Regulation (EU) 2022/850, e-CODEX will provide the technical solution for the justice sector to connect the ICT systems of the competent national authorities, such as the judiciary or other organisations. Through its decentralised infrastructure, e-CODEX allows the electronic exchange of any content transmissible in electronic format, "such as text or sound, visual or audiovisual recordings, in the form of either structured or unstructured data, files or metadata".[91] The e-CODEX system deployed by each Member State consists of two main software components: *(a)* a gateway for the secure exchange of messages with other gateways;[92] and *(b)* a connector that provides many functionalities, including the connection of the national gateway to the national application, the verification of electronic signatures via a security library, ensuring that the messages and attachments cannot be manipulated between the sending connector to the receiving connector, semantic interoperability of the exchanged messages and proof of delivery.

This infrastructure was developed by a broad consortium of Ministries of Justice of Member States, with the support of EU funding, between 2010 and 2016 and is now managed by a consortium of Member States and other organisations, financed by an EU grant (Velicogna and Steigenga 2016). However, given the importance of the e-CODEX system for cross-border exchanges in the area of judicial cooperation in the Union, it is now established by means of an EU legal framework that provides for rules regarding

---

[88] Working party on e-Law (e-justice)—expert group on e-Service of documents and e-Communications Delegations, Brussels, 19 July 2018, 11275/18, p. 17. In any case, it should not be overlooked that since the end user of the service must typically be served on paper, electronic communication may be limited to that between the sending and receiving entities. Electronically transmitted documents and forms must still be printed to be served on the recipient. This could actually raise validity issues regarding the conformity of the document served with the original.

[89] Regulation 2020/1784, Article 5. Cross-border exchanges is carried out without the involvement of any EU institution in processing case data; the Commission is only responsible for the creation, maintenance and future development of a reference implementation software, which Member States may choose to adopt as a back-end system instead of national ones.

[90] E-CODEX is a major project co-funded by the European Union since 2010 that has shown its efficiency in pilots on civil commercial matters.

[91] See Recital 9, Regulation (EU) 2022/850.

[92] At present, the gateway is based on a building block of the Connecting Europe Facility maintained by the Commission known as 'eDelivery', while the management of the connector is carried out by the entity managing the e-CODEX system. See Recital 11, Regulation (EU) 2022/850.

its functioning and development and that ensures the protection of fundamental rights as provided for in the Charter of Fundamental Rights of the EU.[93] Moreover, e-CODEX will be soon handed over to the European Union Agency for the Operational Management of Large-Scale IT Systems in the area of freedom, security and justice (eu-LISA), so as to ensure its long-term sustainability and its governance, while respecting the principle of the independence of the judiciary.

As far as the Service Regulation is concerned, in practice, once all the technical measures have been taken to make this ICT system operational, the transmitting agencies should be able to use their usual national application interface (should one exist) or a software provided by the European Commission (a reference implementation) to send the documents to be notified to the receiving agencies via the e-CODEX system. The specific standard form of the request—which contains information on the nature of the documents and the recipient's own notification—will be completed in electronic format in one of the official languages of the requested State or in a language accepted by that State. The receiving agency, for its part, will send an automatic acknowledgement of receipt to the transmitting agency via the same system, using the electronic version of the forms available in the annex to the Regulation, before notifying the addressee. It goes without saying that relying on e-CODEX means that all documents passing between transmitting and receiving agencies will be able to be signed electronically and will not be deprived of legal effect or considered inadmissible simply because they are in electronic format.

*The Digitalisation of the Notification Procedure and the Impact on National Agencies—A One-Size-Fits-All Solution . . . or, Not?*

The changes brought about by the new Services Regulation have the potential to significantly improve the notification procedure mediated by national agencies. By replacing the traditional paper medium, this system has the potential to reduce the time needed to transmit documents across borders and to eliminate the risks that can arise during postal delivery (e.g., loss of documents, etc.) at all stages of the procedure, including the transmission of the many standard forms that mark the course of the procedure. Also, as emerged in the experience carried out with the European Order for Payment and European Small Claim procedures, the use of e-CODEX platform can favour the coordination with the national rules, since, while its functioning relies on the existing legal, technological and organisational installed base (Hanseth and Lyytinen 2010; Lanzara 2014; Velicogna and Lupo 2017)[94] and the interoperability of domestic e-justice systems (Borsari et al. 2012; Velicogna and Lupo 2017), it supports mutual understanding and the establishment of governance mechanisms (Velicogna and Steigenga 2016).

To fully understand the potential gains of this digitalisation process, however, it is necessary to better explore what impact the new digital procedure will have on the day-to-day work of national transmitting and receiving agencies, which—as a result of the reform of the Services Regulation and the establishment of e-CODEX as a mandatory means of service—become custodians of the entire decentralised electronic transmission system between the Member States. In this respect, the first issue to be addressed is undoubtedly the adaptability of national agencies to work in a fully digital environment. The complete digitalisation of the service procedure to be effective and lead to tangible results requires conditions to be met regarding both the ICT hardware and software available in the offices, both specific digital know-how, which cannot be taken for granted.

As far as Italy is concerned, for example, the responses to the online questionnaires and interviews showed that despite full support for a process of computerisation of the system—which is considered likely to produce a very positive impact in terms of speed and

---

[93]　Regulation (EU) 2022/850 of the European Parliament and of the Council of 30 May 2022 on a computerised system for the cross-border electronic exchange of data in the area of judicial cooperation in civil and criminal matters (e-CODEX system), and amending Regulation (EU) 2018/1726.

[94]　Installed base here refers to the "set of existing technological, legal and organizational components and their "capabilities [. . .] their users, operations and design communities".

certainty—the baseline scenario is not well suited to embrace the change. Notably, UNEP staff point out that, at present, the available resources are by no means adequate to support a rapid adaptation to the digitalisation of the cross-border procedure and that a significant investment in computer equipment would be necessary.[95] In addition, the urgent need for intensive training in the use of ICT tools, which the staff of Italian agencies is currently severely lacking, was unanimously pointed out. Overall, issues related to the training of practitioners were consistently reported as crucial by respondents, not only with regard to IT skills. Most notably, all respondents indicated the need to improve their specific legal and operational skills. The results of the survey revealed, in fact, the need for nationally organised courses on the content of the Regulation, as only a tiny percentage of UNEP staff claim to be familiar with EU legislation.

Furthermore, upgrading language skills should also be regarded as essential, as, at present, the practitioners do not consider the proficiency level sufficient to conduct an increasingly intensive dialogue with foreign authorities. These are far from being minor aspects, today more so than in the past, because they are real enabling factors, in the absence of which the changes brought about by the renewed Service Regulation could go unheeded. Training-related issues, in particular, play an essential role in the smooth running of the procedure and in terms of the sound enforcement of the safeguards provided for in the legislation.[96]

## 6. Study Limitations

This study presents two main limitations to its potential extensibility. The first limitation is that the research described in the paper, while carried out within a broader project involving several EU Member States, focuses on one EU Member State, which limits the possibility of generalising from the results. At the same time, the findings are not limited to the Italian case, as the information provided results from the interaction of Italian experts with legal and organisational requirements, practices, specificities and issues of the other countries involved in the cross-border service of documents.

The second main limitation concerns the timing of the study, as it was carried out before the entry into force of the new Regulation. While the study allowed us to explore the experience of the practical implementation of the Service Regulation (EC) No 1393/2007, which is a valuable achievement per se, it only points to potential issues that may emerge from the implementation of Regulation (EU) 2020/1784 but does not explore them directly. At the same time, the application of the new Regulation is still too recent to provide more than an indication of possible issues to research carried out at the moment of the publication of this study. Furthermore, the provisions referring to the decentralised IT systems based on e-CODEX will apply only after a period of three years after the date of entry into force of the implementing acts the Commission will adopt according to art. 25 of the Regulation (EU) 2020/1784. To address this limit, further research will be required once the recast Regulation is sufficiently experienced by practitioners.

Another limitation could be seen in the number of respondents being limited compared to the number of UNEP personnel. At the same time, considering that UNEP offices have, according to the interviews carried out, at most one officer specialising in cross-border service of documents, the response rate could be considered to be around 28% of the

---

[95] It must be noted that to date there is not even a proper case management system, as the one currently in use at each UNEP (Gestione Servizi UNEP—GSU web) does not cover procedures that fall under the scope of the Service Regulation. These procedures are managed through separate electronic registration systems (in many cases only an excel file), which do not allow for keeping a scanned copy of documents or for keeping proper statistics (e.g., incoming and outgoing flows).

[96] We particularly refer to the right to refuse the document served (now governed by Art. 12), for example, imposes precise duties on the receiving agencies to be observed at the time of service. The person effecting service must be well aware that when the document is not drawn up or is not accompanied by a translation into the official language of the place where service is to be effected, the addressee must be informed of his or her right to refuse service and how to exercise that right. Furthermore, the appropriate form must be delivered in the official language of the Member State of origin and in a language that the addressee understands.

experienced personnel. In other words, a higher number of responses could have reduced the quality and reliability of the provided data. This should also be considered in light of the study's objective, which is to generate a robust understanding of the phenomenon rather than a statistical representation of it (Huberman and Miles 1994; Oppong 2013; Sim et al. 2018; Boddy 2016; Blaikie 2018).

A potential limit could also be seen in the poor performance of the Italian justice system. However, it should be noted that the service of documents in Italy is well-performing and highly digitalised, with the possibility to serve electronically all businesses, professionals and public administrations, which must have a certified e-mail address where they can be served.

## 7. Concluding Remarks

To face the challenges of an ever-integrated society and further improve a genuine area of justice in civil matters, over the last years, the EU institutions have developed a variety of legal instruments to enhance the framework for cross-border judicial cooperation and provide citizens access to justice throughout the EU. This includes specific provisions to make faster, less expensive and more reliable cross-border service of judicial and extrajudicial documents in civil and commercial matters, while safeguarding the recipient's rights.

The Service Regulation is the most important EU legal tool addressing this issue, providing a variety of mechanisms that national actors can use to favour the rapid and successful execution of the service procedure across national borders. In particular, the 'standard' procedure it foresees, which is based on the collaboration between transmitting and receiving Member State agencies, is considered the cornerstone of the European service regime, since it streamlines the procedure, creating a direct channel for cross-border transmission between local intermediaries and removes onerous procedural steps, at least from a legal viewpoint.

While improving the previous situation, Regulation 1393/2007 struggled to sufficiently speed up and make the notification process more efficient, and there are reasons to believe that even the recast Regulation will not be able to achieve this goal fully. Besides the many legislative gaps resulting from the EU's limited competence in this area, which requires constant reference to national rules and procedures, other factors make the notification process generally unsatisfactory. The unclear wording of some provisions, language barriers and a widespread unfamiliarity with EU legislation are some of the main shortcomings identified through the various evaluation exercises and survey-based research conducted so far.

Faced with poor results, a legislative reform of the legal framework has recently been completed. Regulation 2020/1784/EU, in force since July 2022, replaces the current paper-based transmission mechanisms with a decentralised ICT system consisting of national applications interconnected by a secure and reliable communication infrastructure (e-CODEX). Overall, the new framework relies (almost) entirely on broader and better use of technological solutions to increase the efficiency of the cross-border service provision process. The digitalisation of the procedure offers, after all, good opportunities to improve the system in terms of efficiency. The secure exchange of electronic documents between sending and receiving agencies could alleviate some complications arising from the paper procedure, first and foremost reducing notification time and security problems. It would also provide national agencies with a direct and secure channel of communication, including for consultation purposes.

However, the results obtained through the empirical examination of the practices developed in Italy confirm that many challenges lie beyond the successful use of the available technologies. Overall, the organisational structure of Italian offices acting as transmitting and receiving bodies is not designed to adequately support the transmission of notifications at the international level. In particular, except for a few offices with contact persons in charge of liaising with foreign authorities when necessary, there are no specialised units or contact persons in charge of international activities. The staff generally lacks specialisation

in the cross-border service of documents. This is mainly due to the fact that dealing with this procedure is still considered a 'niche market' in Italy, and practitioners are not familiar with the EU Regulation, partly because there is a lack of specialised training in this field and partly because there is no sufficient number of incoming and outgoing procedures to push for local specialisation. Language skills are also not always considered sufficient to adequately perform tasks involving constant contact with foreign jurisdictions and actors. According to the online survey, only a tiny percentage of the staff involved stated they had a very good knowledge of at least one foreign language.

This lack of training undoubtedly affects the application of certain tasks under the Regulation. This is the case with the incorrect use of standard forms, the consultation with foreign authorities, which rarely takes place, or the information to be given to the addressee on the possibility of refusing the documents served based on linguistic criteria, which is not always performed correctly.

It follows that for a digitalised cross-border procedure to be effective, at least two fundamental aspects must be taken into account. First, the establishment and operation of a secure system for the electronic exchange of documents risks being hampered by legal and, above all, practical problems stemming from the existence of different regulatory levels—European, national, and local—that the new regulation does not address. The decentralised nature of the system to be adopted and the flexibility in the choice of implementation software to be applied locally will certainly help to respect these legal levels and support the exchange of legally valid communications. However, technology alone cannot overcome all the obstacles arising from the need to coordinate EU rules with national procedures, the proliferation of local practices, or concerns about protecting certain procedural rights when moving from paper-based to digital exchange.[97]

The same can be said about the variety of practical problems that arise from everyday practice and are mostly due to a lack of guidance on how to apply EU rules in a concrete case and a general lack of familiarity with the EU service regime. One only has to think of the difficulties in correctly filling in the standard forms to be used, the translation problems, the gaps in the rules to be followed to ensure that on the one hand, the addressee's right to refuse the document is effective, and on the other, it is not abused to the detriment of the applicant's legitimate interests. Moreover, the digitalisation of the cross-border procedure itself may not be smooth, as the availability of adequate computer equipment cannot be given for sure, and the improvement of the staff's computer skills must also be addressed.

Finally, it is worth emphasising that, just like many other cross-border procedures, the transmission mechanism established by the Services Regulation relies heavily on the principle of cooperation and mutual acceptance between national actors. It follows that, beyond the available state-of-the-art ICT tools, which certainly play a key role, this process can only really work in a cross-border working environment where a level playing field is guaranteed, and local actors can connect with their counterparts effectively. These results can be achieved only with a significant investment in training initiatives to make practitioners aware of existing legal possibilities and to improve their skills and abilities, so they can use them.

Although the paper focused on the Italian case, where we carried out an in-depth investigation of formal rules and actual practices, the literature on the subject and the

---

[97] Cft. Steigenga E., Taal, S., Medici, A., and Velicogna, M., Pro-CODEX Report Exploring the potential for a Service of Documents e-CODEX use case in The Netherlands, finalised 12 June 2018. This Report, focused on the Dutch system, shows how the possibile digitalisation of the procedure could affect certain procedural rights of the addressee, notably the personal verification of the acknowledgment of receipt. In paper-based procedure, this step is carried out by bailiffs, who deliver the documents in person to litigants. However, the digital personal acknowledgement is currently not possible. This results in a lack of assurance that the addressee has received and is aware that the document has been served on him/her. The Report also stresses that, at present, certified e-mail service or infrastructure are not available in the Country. Please note that this Report has been realised within the framework of "Pro-CODEX: Connecting legal practitioners' national applications with e-CODEX infrastructure", project co-funded by the European Commission Directorate-General Justice within the Justice Programme (2014–2020), Action Grant to support judicial cooperation in civil matters Application: JUST/2014/JCOO/AG/CIVI 4000007757.

evaluations carried out by the European institutions show that the elements we highlighted are present, to varying degrees, in all Member States and should therefore be addressed in the future.

**Author Contributions:** Conceptualisation, R.A. and M.V.; Data curation, R.A. and M.V.; Formal analysis, R.A. and M.V.; Funding acquisition, M.V.; Investigation, R.A. and M.V.; Methodology, R.A. and M.V.; Project administration, M.V.; Resources, M.V.; Software, R.A. and M.V.; Supervision, R.A. and M.V.; Validation, R.A. and M.V.; Visualisation, R.A. and M.V.; Writing—original draft, R.A. and M.V.; Writing—review and editing, R.A. and M.V. All authors have read and agreed to the published version of the manuscript.

**Funding:** This research was funded by Justice Programme of the European Union "Me-CODEX II: Maintenance of e-Justice Communication via Online Data Exchange": JUST/CEF-TC-2018-CSP-ECODEX.

**Institutional Review Board Statement:** The study was conducted according to the guidelines of the CNR (Consiglio Nazionale delle Ricerche) ethical committee. Given that the qualitative study involved a limited number of observations and that data gathering operations did not imply risks for data protection, human dignity and health and bioethics, the ethical approval was not necessary.

**Informed Consent Statement:** Informed consent was obtained from all subjects involved in the study. For further details please see the "Background and methodology section".

**Data Availability Statement:** This study is based on quantitative and qualitative data gathering through online surveys and semi- structured interviews. Data are described within the paper and not available in original form for privacy reasons.

**Conflicts of Interest:** The authors declare no conflict of interest.

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
