# Peer review of "Cross-Border Document Service Procedures in the EU from the Perspective of Italian Practitioners—The Lessons Learnt and the Process of Digitalisation of the Procedure through e-CODEX"

_laws, 2022_

Round 1

Reviewer 2 Report

·      

The piece is an extremely interesting contribution to civil procedure developments and the studied conducted is both laudable and worthwhile. Suggestions below are for the editing of the piece:

   Shorten the title significantly- The overall focus was not clear- clarify what is main subject

The piece needed to be shortened significantly

·         The instrument is alleged to be highly ineffective yet is under review and the subject of legislative change- this presents itself as exploratory empirical work but needs to be clearer on how links into official reviews etc- more work was needed on its originality and relevance as a result- also some further analysis of the impact assessments and how the challenges outlined here mirror those in EU wide reviews

·         Could be situated a bit more clearly in relevant literature

·         Emphasise more the practical problems resulting from the instrument

·         Outline the impact assessment further

·         Some competence issues are hinted at, also the reason for the introduction of the instrument could be outlined with more clarity

·         Could be clearer the perspective adopted - summarising a range of ‘practitioner views’- and analysing this- what about academia, foreign perspectives, was too introspective in places

·         National ICT issues were possibly problematic and needed more deference and explanation

·         Is anyone litigating this/ writing about it- try to reference further on this more clearly at the outset

·         Strengthening national agencies appears as a key result of this but this is rather buried. How is this strengthening weakened by what is outlined, ie make the problems resulting clearer for the reader

Reviewer 3 Report

This is an interesting article and its originality lies in the underlying empirical research conducted in Italy. Congratulations to the author on that empirical work. It is important that we learn more about how the EU civil justice regulations work in practice in the Member State.

That being said the response rate and number of interviews was relatively low. This is mentioned in Section 2 outlining the methodology. However, one should perhaps also mention this in the analysis/conclusions, i.e. it is good to be transparent that the results cannot be considered conclusive rather are indicative.  (See also comment in the attached document).

In relation to structure, I note that there is a clear structure and good overall flow. Some minor repetitions though in the text. In relation to format issues, I have one comment on a short paragraph  and one comment on sub-headings (see attached document).

In relation to footnotes and use of sources I note the following: 

- The technique with abbreviated names for central sources and cross-referencing is not used, which in my opinion make the footnotes "heavy" to read.  (However, this may be preference of the journal.)

-  I note that when using the EU-Commission documents, specific page numbers are not set-out in several places where one would expect a page number, as examples see notes 30 and 31.

- I also note that quite general statements are made in the text, for example in Chapter 1, about the application of the Service Regulation in general in Europe and the source used is the Commission Report from 2013, but that is nearly 10 years old. It may not be an accurate depiction of the application of the Regulation in all Member States of the EU today. Unless there are more up-to-date sources, I would suggest being a bit more cautious in the formulation of the statements made. 

- In some places I miss a footnote completely, where I would expect a footnote. I have highlighted some in the attached document. 

- When explaining the "standard" system of service under the Regulation and the different phases, in particular on page 9, I would have expected mention of the relevant articles in the Regulation either in the text itself or in the footnotes. 

- I am also missing a few articles from international scholarship that could support and deepen the analysis and discussion  in sections 5 and 6. For example by  Gascon Inchausti, Fernando in the Journal of private international law 09/2017. In related to both Service, EU  civil justice and digital development. 

- There are a few miss-spellings or incorrect formulations in document due to fact that the author is a non-native speaker. I have highlighted a few linguistic issues/questions in the attached PDF-document. 

Finally, the analysis in the article is fairly good. However, it could have been deeper in particular in chapters 5 and 6. 

Round 2

Reviewer 1 Report

thank you for addressing the comments